# A surge in respiratory syncytial virus infection-related hospitalizations associated with the COVID-19 pandemic: An observational study at pediatric emergency referral hospitals in Tokushima Prefecture

**Koichi Shichijo** [1]*, **Shoko Fukura**[1], **Shunsuke Takeuchi**[1], **Takahiro Tayama**[1], **Akemi Ono**[1], **Yuko Ichihara** [1], **Kenichi Suga** [1], **Hiroki Sato**[2], **Atsumi Takechi**[2], **Sonomi Matsumoto**[2], **Shuji Fujino**[2], **Takako Taniguchi** [2], **Akiyoshi Takahashi**[2], **Tsutomu Watanabe** [2], **Shuji Kondo** [1]

**1** Department of Pediatrics, Tokushima Prefectural Central Hospital, Tokushima, Japan, **2** Department of Pediatrics, Tokushima Red Cross Hospital, Komatsushima, Japan

* kshichijo.32@gmail.com

## Abstract

The outbreak of coronavirus disease (COVID-19) resulted in implementation of social distancing and other public health measures to control the spread of infection and improve prevention, resulting in a decrease in respiratory syncytial virus (RSV) and pediatric respiratory tract infection rates. However, there was a rapid and large re-emergence of RSV infection in Japan. Notably, we were faced with a difficult situation wherein there was a shortage of hospital beds. This study aimed to examine the epidemiological patterns of RSV-related hospitalizations among children before and after the COVID-19 pandemic onset at two pediatric emergency referral hospitals covering the entire Tokushima Prefecture. Data were extracted from electronic medical records of children hospitalized with RSV infection between January 1, 2018, and December 31, 2021. All patients meeting the eligibility criteria were included in this study. The rates of study outcomes were documented annually during 2018–2021 and compared between the 2018–2020 and 2021 periods. In 2020, there was no RSV infection outbreak. Hospitalizations at the peak week in 2021 were 2.2- and 2.8-fold higher than those in 2018 and 2019, respectively. Hospitalizations in 2021 were concentrated within a short period. In addition, there was a significant increase in hospitalizations among children aged 3–5 months and those older than 24 months. The high-flow nasal cannula (HFNC) use rate nearly doubled in 2021. A new pandemic in the future may cause an outbreak of RSV infection that can result in an intensive increase in the number of hospitalizations of pediatric patients requiring respiratory support, especially in infants aged <6 months. There is an urgent need to improve the preparedness of medical systems, particularly in terms of the number of inpatient beds and the immediate availability of HFNC.

**Data Availability Statement:** All relevant data are within the paper and Supporting Information files.

**Funding:** The authors received no specific funding for this work.

**Competing interests:** The authors have declared that no competing interests exist.

## Introduction

The outbreak of coronavirus disease (COVID-19) resulted in the worldwide implementation of social distancing and other public health measures to control spread of infection and improve prevention of the disease. In Japan, the public was encouraged to adopt infection control measures, such as face mask use and hand washing, and avoid the "three Cs" (closed spaces, crowded places, and cross-contact settings) via physical distancing [1]. These measures resulted in a decrease in the rates of pediatric respiratory infections, including respiratory syncytial virus (RSV) infections [2]. RSV infections affect children aged <2 years and can cause severe disease in infants aged <6 months [3]. There were an estimated 33 million cases of RSV-associated lower respiratory tract infections among children aged 0–60 months worldwide in 2019, including 3.6 million hospitalizations and 26,300 deaths [4].

Following the relaxation of COVID-19 measures, a re-emergence of RSV infections was observed worldwide [5–8]. In contrast, even in Japan, where infection control measures continue, the number of cases has increased nationwide since March 2021, with incidence rates peaking in July [9], including in Tokushima Prefecture [10]. At our hospitals, the number of hospitalized RSV infection patients increased massively. We were faced with a difficult situation wherein there was a shortage of hospital beds and high-flow nasal cannulas (HFNCs). The major change in the epidemiological pattern of RSV infection transmission observed in 2021, compared with that in previous years, was the increase in the age at infection among children, with those aged ≥2 years being more susceptible to infection [11]. However, it has not been well investigated whether infection numbers and severity, as well as age at infection, have changed and how these changes have impacted medical care for hospitalized children.

This study aimed to compare the patterns of pediatric RSV-related hospitalizations between the period before and after the onset of the COVID-19 pandemic at two pediatric emergency referral hospitals in Tokushima Prefecture. These two hospitals provide pediatric care for the entire Tokushima Prefecture; therefore, the presented findings on RSV hospitalizations can be considered to provide an overall representation of the region.

## Materials and methods

### Ethics approval

This study was performed in line with the principles of the 1964 Declaration of Helsinki and its later amendments. Approval was granted by the Tokushima Prefectural Central Hospital Ethics Committee (approval number 21–37) and Tokushima Red Cross Hospital Ethics Committee (approval number 399). The requirement for informed consent was waived because the study was a retrospective study.

### Study design and patients

We conducted a retrospective observational study of all pediatric patients admitted to the two referral hospitals in Tokushima Prefecture; data were extracted from nominal identification of electronic medical records of children hospitalized with RSV infection before and after the onset of the COVID-19 pandemic. Children who were admitted to the Department of Pediatrics at Tokushima Prefectural Central Hospital (public funding) and the Department of Pediatrics at Tokushima Red Cross Hospital (public funding), in Tokushima Prefecture, Japan, between January 1, 2018, and December 31, 2021, and who met the following eligibility criteria (1 or 2) were included in this study: entry in the electronic medical record as any of the following–"RSV infection," "RSV bronchiolitis," "RSV pneumonia," and "RSV encephalopathy"; and treated on site during the study period, as evidenced by medical records.

1. positive results on a rapid antigen test

2. clinical diagnosis of RSV infection by a physician, with an outbreak of RSV infection in the surroundings (including negative results on a rapid antigen test or without the test)

Cases without an outbreak of RSV infection in the surroundings and negative results on a rapid antigen test were excluded.

## Data source and collection

Tokushima Prefecture is located in the eastern part of Shikoku island, measuring 107.37 km from east to west and 79.03 km from north to south. It is connected to Honshu island by a bridge, accessible within a day trip from Osaka city [12]. In October 2020, the general population in Tokushima Prefecture was 719,559 and the number of children aged under five years was 22,651 [13]. The study period overlapped with the change in pediatric emergency medical care system schedules of the hospitals. Both participating hospitals had a 24-hour operating pediatric care system that accepted secondary and tertiary emergency patients after hours from 17:00 to 8:30 the following day, as well as primary emergency patients from 22:30 to 8:30 the following day. However, in November 2019, owing to a shortage of pediatricians, the two hospitals introduced a rotating system for providing out-of-hours pediatric emergency care. In July 2020, the Tokushima Prefectural Central Hospital resumed the previous 24-hour emergency care system. During the period of the present analysis (2018–2021), only these two hospitals in Tokushima Prefecture provided out-of-hours emergency care and pediatric emergency hospitalization. In the study, there was no reallocation of pediatric care to the reference hospitals during this period. Therefore, the data obtained from the participating hospitals represent all RSV infection cases recorded during the study period in this area.

Data on the following variables were extracted and reviewed as far back as possible from the non-nominal identification databases of the participating hospitals: number of children hospitalized for RSV infection per week; patient characteristics, including age (months), sex, palivizumab indication (at the season), palivizumab prophylaxis (at the season), presence of siblings; hospitalization duration (days), oxygen use rate (%, [number of children on oxygen/number of hospitalized children] × 100); HFNC use rate ([HFNC children/admitted children] × 100); ventilator use rate (%, [number of ventilator-controlled children/number of hospitalized children] × 100); and mortality rate ([number of deaths/number of hospitalized children] × 100).

Palivizumab indications included: infants aged ≤12 months with a gestational age at birth of ≤28 weeks; infants aged ≤6 months with a gestational age at birth of 29–35 weeks; infants aged ≤24 months who had been treated for bronchopulmonary dysplasia within the previous 6 months; infants aged ≤24 months with congenital heart disease, immunodeficiency, and/or Down syndrome.

## Data analysis

The rates of study outcomes were considered annually during 2018–2021 and were compared between 2018–2020 and 2021. If there was an increase in hospitalization frequency in any age group, we conducted age-stratified analyses. All statistical analyses were performed with EZR (Version 1.40, Saitama Medical Center, Jichi Medical University, Saitama, Japan) [14], which is a graphical user interface for R (The R Foundation for Statistical Computing, Vienna, Austria). More precisely, it is a modified version of R commander designed to add statistical functions frequently used in biostatistics. The Mann–Whitney U test was used to compare age differences. The Welch t-test was used to compare hospitalization duration. The Pearson chi-

square test was used for all other comparisons. *P*-values <0.05 indicated a statistically significant difference.

## Results

In this study, 834 cases were analyzed. Out of the total 834 cases, positive results on a rapid antigen test comprised 820 (98.3%), and clinical diagnosis of RSV infection by a physician comprised 14 (1.7%). There was no difference in the guidelines for rapid diagnostic testing and clinical diagnosis of RSV infection between the periods. We did not perform the rapid molecular diagnostic test or virus isolation for RSV infections. The maximum age of children was 172 months. Among the 834 cases, 813 (97.5%) were residents of Tokushima Prefecture, whereas 21 (2.5%) were non-residents. In 2018, the number of children hospitalized with RSV infection peaked at week 35 (27 August to 2 September), with 23 new hospitalizations/week (Fig 1). In 2019, the peak was at week 38 (16–22 September), with 18 new hospitalizations/week. In 2020, there were no infection peaks. In 2021, no hospitalizations were recorded before week 18; however, the number of hospitalizations peaked at week 30 (26 July to 1 August), with a rate of 50 hospitalizations/week; this represented 2.2- and 2.8-fold increases relative to values recorded in 2018 and 2019, respectively. To assess the degree of hospitalization during the epidemic, we examined the percentages of annual hospitalizations in the 3 weeks before and after the epidemic peak, which were 40.0% (107/267) in 2018, 29.8% (85/285) in 2019, and 86.3% (227/263) in 2021, indicating that significantly more hospitalizations were confined to a shorter period of time (both *P*<0.001).

The annual number of hospitalizations was higher in 2018 (n = 267) and 2019 (n = 285) than in 2020 (n = 19) and comparable with that in 2021 (n = 263) (Table 1). Among children hospitalized for RSV infection, the proportion of those aged <6 months and ≥24 months increased in 2021 compared to that during 2018–2020. Specifically, there was a significant increase in the percentage of hospitalizations among children aged 3–5 months (11.6 vs. 16.7%; *P*<0.05) and among those aged ≥24 months (14.9 vs. 24.7%; *P*<0.001) in 2021. In

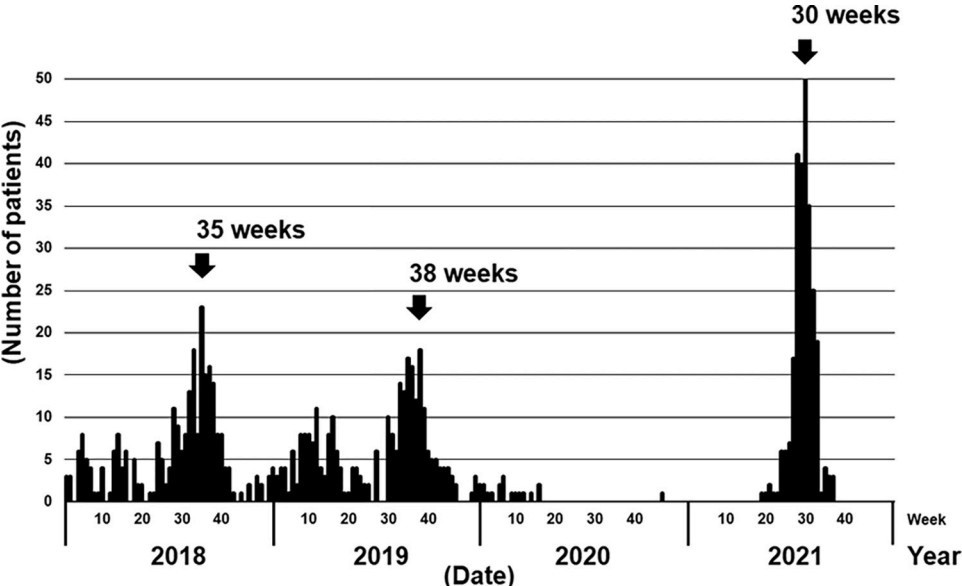

**Fig 1. Number of children hospitalized due to RSV infection per week.** RSV infections occurred throughout the year, with a peak arising before the COVID-19 pandemic. After the onset of the COVID-19 pandemic, RSV infection was not observed outside the peak season. COVID-19, coronavirus disease; RSV, respiratory syncytial virus.

**Table 1. Annual changes in characteristics of children hospitalized with an RSV infection.**

|  | 2018 (n = 267) | 2019 (n = 285) | 2020 (n = 19) | 2021 (n = 263) | *P*-value |
|---|---|---|---|---|---|
| **Age: months, n (%)** |  |  |  |  |  |
| <3 months | 77 (28.8) | 70 (24.6) | 3 (15.8) | 73 (27.8) | 0.674 |
| 3–5 months | 30 (11.2) | 33 (11.6) | 3 (15.8) | 44 (16.7) | **0.0472** |
| 6–11 months | 39 (14.6) | 45 (15.8) | 5 (26.3) | 31 (11.8) | 0.168 |
| 12–23 months | 87 (32.6) | 89 (31.2) | 5 (26.3) | 50 (19.0) | **0.000123** |
| ≥24 months | 34 (12.7) | 48 (16.8) | 3 (15.8) | 65 (24.7) | **0.000919** |
| **Sex: male, n (%)** | 153 (57.3) | 153 (53.7) | 10 (52.6) | 142 (54.0) | 0.765 |
| **Palivizumab indication** |  |  |  |  |  |
| n (%) | 8(3.0) | 4 (1.4) | 0 (0) | 2 (0.8) | 0.246 |
| not indicated | 259 (97.0) | 281 (98.6) | 19 (100.0) | 261 (99.2) |  |
| **Palivizumab prophylaxis** |  |  |  |  |  |
| n (%) | 5 (1.9) | 2 (0.7) | 0 (0) | 0 (0) | 0.105 |
| not administered | 262 (98.1) | 283 (99.3) | 19 (100.0) | 263 (100.0) |  |
| **Presence of siblings** |  |  |  |  |  |
| n (%) | 151 (56.6) | 164 (57.5) | 10 (52.6) | 187 (71.1) | **0.0154** |
| no siblings | 45 (16.9) | 44 (15.4) | 4 (21.1) | 31 (11.8) |  |
| unknown | 71 (26.6) | 77 (27.0) | 5 (26.3) | 45 (17.1) |  |
| **Hospitalization duration** |  |  |  |  |  |
| mean days (SD) | 5.31 (1.97) | 5.50 (2.17) | 5.58 (1.61) | 5.52 (2.50) | 0.535 |
| **Oxygen use rate, n (%)** | 107 (40.1) | 122 (42.8) | 6 (31.6) | 122 (46.4) | 0.175 |
| **HFNC use rate, n (%)** | 17 (6.4) | 25 (8.8) | 2 (10.5) | 36 (13.7) | **0.00784** |
| **Ventilator use rate, n (%)** | 2 (0.7) | 1 (0.4) | 0 (0) | 3 (1.1) | 0.386 |
| **Mortality rate, n (%)** | 0 (0) | 0 (0) | 0 (0) | 1 (0.4) | NA |

HFNC, high-flow nasal cannula; RSV, respiratory syncytial virus; SD, standard deviation

contrast, the percentage of hospitalizations among children aged 12–23 months decreased in 2021 compared to 2018–2020 (31.7 vs. 19.0%; *P*<0.001). There was no significant change in palivizumab indication (2.6 vs. 2.3%). Prophylaxis rates among palivizumab-eligible children were 70% (7/10) in 2018, 60% (3/5) in 2019, and 66% (2/3, excluding three cases with an unknown status) in 2021. Among children who had siblings at home, a greater proportion was hospitalized in 2021 than in 2018–2020 (56.9 vs. 71.1%; *P*<0.05). The duration of hospitalization did not change (5.42 vs. 5.52 days). Oxygen use rates increased slightly (41.2 vs. 46.4%) and HFNC use rates increased significantly (7.7 vs. 13.7%; *P*<0.01). Ventilator use rates were low in 2018–2020 (0.5%) and 2021 (1.1%). Only one RSV-related death was recorded in 2021, which was due to cardiopulmonary arrest on arrival.

Increases in hospitalizations among children aged <6 months and ≥24 months were examined separately. Among infants aged <6 months, there was a significant increase in hospitalizations among those who had siblings at home in 2021 (74.1% vs. 94.0%; *P*<0.01) (Table 2). The use of HFNC also increased significantly in 2021 (16.7% vs. 26.5%; *P*<0.05). Among children aged ≥24 months, there were no significant differences in either the presence of siblings or HFNC use in 2021 (Table 3).

## Discussion

The present study highlights that a rapid and large increase in the number of hospitalizations within a short period resulted in a shortage of hospital beds. As a temporary measure, we

**Table 2. Epidemiological changes in characteristics of infants aged <6 months.**

| | 2018–2020 (n = 216) | 2021 (n = 117) | P-value |
|---|---|---|---|
| **Sex: male, n (%)** | 117 (54.2) | 59 (50.4) | 0.566 |
| **Palivizumab indication, n (%)** | 8 (3.7) | 1 (0.9) | 0.168 |
| not indicated | 208 (96.3) | 116 (99.1) | |
| **Palivizumab prophylaxis, n (%)** | 6 (2.8) | 0 (0) | 0.0944 |
| not administered | 210 (97.2) | 117 (100.0) | |
| **Presence of siblings, n (%)** | 160 (74.1) | 110 (94.0) | **0.00256** |
| no siblings | 20 (9.3) | 2 (1.7) | |
| unknown | 36 (16.7) | 5 (4.3) | |
| **Hospitalization duration** | | | |
| mean days (SD) | 6.05 (2.38) | 6.37 (2.98) | 0.316 |
| **Oxygen use rate, n (%)** | 100 (46.3) | 63 (53.8) | 0.207 |
| **HFNC use rate, n (%)** | 36 (16.7) | 31 (26.5) | **0.0444** |
| **Ventilator use rate, n (%)** | 3 (1.4) | 2 (1.7) | 1 |
| **Mortality rate, n (%)** | 0 (0) | 0 (0) | NA |

HFNC, high-flow nasal cannula; SD, standard deviation

managed to increase the number of pediatric beds from 20 to 30 at the Tokushima Prefectural Central Hospital. However, we could not admit children who needed hospitalization, even if they were only a few months old. We had to follow them up carefully and frequently as outpatients every day.

In addition, there was an increase in hospitalizations of children aged <6 months. Most RSV-related deaths worldwide occur in children aged 0–6 months [4]. Due to the hospital bed shortage, vulnerable infants aged <6 months were given priority hospitalization. Additional HFNC therapies were selected for them because HFNC is useful in mild to moderate cases of bronchiolitis that do not improve with oxygen therapy [15]. However, such therapies could not be provided to all children who required it, as both hospitals faced a shortage of HFNCs.

**Table 3. Epidemiological changes in characteristics of children aged ≥24 months.**

| | 2018–2020 (n = 85) | 2021 (n = 65) | P-value |
|---|---|---|---|
| **Sex: male, n (%)** | 40 (47.1) | 39 (60.0) | 0.138 |
| **Palivizumab indication, n (%)** | 0 (0) | 0 (0) | 1 |
| not indicated | 85 (100.0) | 65 (100.0) | |
| **Palivizumab prophylaxis, n (%)** | 0 (0) | 0 (0) | 1 |
| not administered | 85 (100.0) | 65 (100.0) | |
| **Presence of siblings, n (%)** | 39 (45.9) | 29 (44.6) | 0.833 |
| no siblings | 18 (21.2) | 15 (23.1) | |
| unknown | 28 (32.9) | 21 (32.3) | |
| **Hospitalization duration** | | | |
| mean days (SD) | 5.04 (1.71) | 4.80 (1.88) | 0.431 |
| **Oxygen use rate, n (%)** | 24 (28.2) | 25 (38.5) | 0.22 |
| **HFNC use rate, n (%)** | 3 (3.5) | 3 (4.6) | 1 |
| **Ventilator use rate, n (%)** | 0 (0) | 0 (0) | NA |
| **Mortality rate, n (%)** | 0 (0) | 0 (0) | NA |

HFNC, high-flow nasal cannula; SD, standard deviation

This emphasizes the need to prepare a sufficient reserve supply of HFNCs for pandemic events.

We also found an increase in hospitalizations of children older than 24 months in 2021, the details of which are shown in Table 3. The data reflect the increase in the number of children who were susceptible to RSV infection. The need for oxygen use in these children increased by only 10%. This means that the RSV severity in children aged ≥24 months was not so high. The percentage of children aged ≥24 months who had siblings at home did not differ between 2018–2020 and 2021, whereas the corresponding percentage in infants aged <6 months was higher in 2021. This might explain one of the reasons for younger infants to get infected at home by older siblings during the COVID-19 pandemic.

In Finland, the re-emergence of RSV infections has been associated with the lifting of COVID-19-related restrictions [7]. Therein, a resurgence of influenza has occurred; however, an influenza outbreak has not been observed in Japan yet [16], suggesting that the epidemiology of infectious diseases after the onset of COVID-19 is different in both countries. The number of COVID-19 cases in Japan was 230,304 (3,414 deaths) on December 31, 2020, increasing to 1,733,325 (18,393 deaths) in 2021 [17–19]. In Tokushima Prefecture, there were 199 cases of COVID-19 in 2020, increasing to a cumulative number of 3,291 cases by the end of 2021 [20], suggesting the outbreak continued beyond 2021. In Japan, in 2021, COVID-19 was centered on socially active adults aged 20–59 years, with few infections among those aged ≤19 years [21]. People with older relatives or young children in their families were more likely to fear the severity of the disease. To date, the Japanese government continues to recommend public health infection control measures, including avoiding crowded places [21]. Although public health measures in Japan remained relatively strict, young children were generally exempt from wearing masks, making them susceptible to RSV infection, which may be acquired at daycare centers and elsewhere. Although age-based infection patterns changed in Japan in 2021, they did not change in Israel [6]. While Japan and Australia observed peak RSV infection outbreaks [8, 22], France did not [5]. Overall, these differences in RSV infection patterns may have been affected by differences in public health measures introduced to control COVID-19 spread, as well as local lifestyles.

The earlier-than-usual outbreak of RSV infections may be because palivizumab was not administered in time. Palivizumab helps to prevent severe disease and complications associated with RSV infection in high-risk children, and hence, its administration to eligible children is strongly recommended [23]. In 2021, there was no significant change in both palivizumab indication and palivizumab prophylaxis among hospitalized children. The Tokushima Pediatric Association alerted local medical institutions regarding the transmission of RSV and provided information based on the National Epidemiological Surveillance of Infectious Diseases in Japan. This may have suppressed RSV infection in high-risk children by facilitating the earlier initiation of palivizumab programs and timely inoculation.

This study had some limitations. Although clinical diagnoses accounted for a minority of cases (1.7% of the total), there is a possibility of incorrect classification due to potential overlap with other respiratory infections and co-infections, as rapid molecular diagnostic test or viral isolation was not performed. Decisions regarding hospitalizations were at the physicians' discretion and depended on inpatient bed availability and family circumstances. In addition, selection of therapies such as HFNC was not only decided at conferences by several doctors, but also by the doctor-in-charge. While this study only presented epidemiological data on RSV infections for a specific region of Japan, the findings may help develop preparedness strategies for future outbreaks.

The number of RSV infection-related hospitalizations increased in children aged <6 months and in those older than 24 months at two pediatric emergency referral hospitals in

Tokushima Prefecture between 2018–2020 and 2021. In 2021, the RSV infection rate peaks were observed 1–2 months earlier than those in the previous years. Hospitalizations were concentrated within a short period of time. Although the current pandemic is not yet over, we need to be aware that such pandemics can cause outbreaks of RSV infection that can increase the number of hospitalizations of pediatric patients requiring pulmonary support. The preparedness of medical systems, particularly in terms of the number of inpatient beds and the immediate availability of HFNCs, needs to be improved in order to handle such emergency situations.

## Supporting information

**S1 Data. Raw data in this study.**
(XLSX)

**S1 Table. Epidemiological changes in characteristics of infants aged 6–11 months.**
(DOCX)

**S2 Table. Epidemiological changes in characteristics of infants aged 12–23 months.**
(DOCX)

## Acknowledgments

We would like to thank Editage (www.editage.com) for English language editing.

## Author Contributions

**Conceptualization:** Koichi Shichijo, Shuji Kondo.

**Formal analysis:** Koichi Shichijo, Shoko Fukura, Shunsuke Takeuchi, Takahiro Tayama, Akemi Ono, Yuko Ichihara, Kenichi Suga, Hiroki Sato, Atsumi Takechi, Sonomi Matsumoto, Shuji Fujino, Takako Taniguchi, Akiyoshi Takahashi, Tsutomu Watanabe, Shuji Kondo.

**Investigation:** Koichi Shichijo, Shoko Fukura, Shunsuke Takeuchi, Takahiro Tayama, Akemi Ono, Yuko Ichihara, Kenichi Suga, Hiroki Sato, Atsumi Takechi, Sonomi Matsumoto, Shuji Fujino, Takako Taniguchi, Akiyoshi Takahashi, Tsutomu Watanabe, Shuji Kondo.

**Methodology:** Koichi Shichijo, Akiyoshi Takahashi, Tsutomu Watanabe, Shuji Kondo.

**Resources:** Shoko Fukura, Shunsuke Takeuchi, Takahiro Tayama, Akemi Ono, Yuko Ichihara, Hiroki Sato, Atsumi Takechi, Sonomi Matsumoto, Shuji Fujino, Takako Taniguchi.

**Supervision:** Tsutomu Watanabe, Shuji Kondo.

**Writing – original draft:** Koichi Shichijo.

**Writing – review & editing:** Kenichi Suga, Akiyoshi Takahashi, Tsutomu Watanabe, Shuji Kondo.

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
