## [Decision Letter · Decision Letter 0]

18 Apr 2023

PGPH-D-23-00225

A surge in respiratory syncytial virus infection-related hospitalizations associated with the COVID-19 pandemic: An observational study at pediatric emergency referral hospitals in Tokushima Prefecture

Dear Dr. Shichijo,

Thank you for submitting your manuscript to PLOS Global Public Health. After careful consideration, we feel that it has merit but does not fully meet PLOS Global Public Health’s publication criteria as it currently stands. Therefore, we invite you to submit a revised version of the manuscript that addresses the points raised during the review process.

We look forward to receiving your revised manuscript.

Kind regards,

Everton Falcão de Oliveira, Ph.D

Academic Editor

Journal Requirements:

2. We ask that a manuscript source file is provided at Revision. Please upload your manuscript file as a .doc, .docx, .rtf or .tex.

3. In the online submission form, you indicated that "Raw data were generated at Tokushima Prefectural Central Hospital and Tokushima Red Cross Hospital. Derived data supporting the findings of this study are available from the corresponding author KS on request". All PLOS journals now require all data underlying the findings described in their manuscript to be freely available to other researchers, either 1. In a public repository, 2. Within the manuscript itself, or 3. Uploaded as supplementary information.

Additional Editor Comments (if provided):

Reviewers' comments:

Reviewer's Responses to Questions

**Comments to the Author**

1. Does this manuscript meet PLOS Global Public Health’s publication criteria? Is the manuscript technically sound, and do the data support the conclusions? The manuscript must describe methodologically and ethically rigorous research with conclusions that are appropriately drawn based on the data presented.

Reviewer #1: Yes

Reviewer #2: Yes

2. Has the statistical analysis been performed appropriately and rigorously?

Reviewer #1: Yes

Reviewer #2: Yes

3. Have the authors made all data underlying the findings in their manuscript fully available (please refer to the Data Availability Statement at the start of the manuscript PDF file)?

Reviewer #1: Yes

Reviewer #2: Yes

4. Is the manuscript presented in an intelligible fashion and written in standard English?

Reviewer #1: Yes

Reviewer #2: Yes

5. Review Comments to the Author

Reviewer #1: This article represents a retrospective study on respiratory syncytial virus infections in a community in Japan (Tokushima prefecture), comparing the pre-pandemic period with the period during/immediately after the COVID-19 pandemic. Overall, it is an interesting study as it represents something that many medical communities have also observed: a possible increase in the number of RSV infections in a short period of time, coinciding with the relaxation of non-pharmacological measures to control COVID-19. Transforming our experiences into data that can be analyzed and bringing practical lessons that can guide public policies and strategic planning is essential.

However, some points deserve to be highlighted. The first point, which I consider crucial for decision-making regarding the publication of the work, refers to the inclusion criteria for the selected patients. On page 6, line 78, the authors state, "The following eligibility criteria were included in this study: positive results on a rapid diagnostic test or clinical diagnosis of RSV infection by a physician." I understand that many paediatricians may give a clinical diagnosis of bronchiolitis, but not all these cases may be caused by RSV. The authors do not report in the results how many diagnoses were made by rapid tests and how many were made clinically. I would like the authors to clarify the criteria used for exclusive clinical diagnosis and what proportion of clinical or laboratory diagnoses were made, as I believe this could impact the results. Another question in this field is about the performance of molecular tests, which are widely used today and were not mentioned in the text. Does the Tokushima prefecture perform molecular tests for patients with bronchiolitis symptoms and negative rapid tests for RSV? Has any other important etiological agent been identified?

Continuing the text, the rest of the methodology, data collection, and statistical analyses seem adequate. The results are presented clearly. Figure 1 is appropriate and clearly emphasizes what the authors want to highlight. The tables are also clear. Only a few adjustments need to be made. On page 12, at the end of Table 1, the Ventilator use rate, n (%) was not statistically significant and is in bold, which needs to be fixed by removing the bold formatting.

Regarding the discussion, the first idea presented is the increase in the number of cases in children under six months (and the increase in the need for oxygen) and also the increase in cases in children over 24 months. The exposure is appropriate, and the evidence of increased infection in children with siblings is very interesting.

On page 16, in the second paragraph, the author presents data on the significant increase in COVID-19 infections in Tokushima comparing 2020 and 2021 and states that the government continues to recommend infection control measures but does not discuss what led to this increase. Tokushima went from 199 cases in 2020 to over 3000 in 2021, and this increase is very significant! Discussing some factors that led to this significant increase is essential to understand the increase in COVID-19 cases and those of RSV.

The data support the conclusions, and the authors present the study's limitations, which I believe are adequate. Overall, it is an interesting article that can be considered for publication, provided that the authors can clarify the raised points and add some epidemiological discussion on the factors that led to the increase in the circulation of respiratory viruses in 2021. Congratulations on the work, and I will be available to contribute if necessary and relevant.

Reviewer #2: The study “A surge in respiratory syncytial virus infection-related hospitalizations associated with the COVID-19 pandemic: An observational study at pediatric emergency referral hospitals in Tokushima Prefecture” proposes to compare the number of hospitalizations due to RSV in children with before and after design, considering the Covid-19 pandemic. Secondary data from two reference hospitals for emergency care of the pediatric population are used. It is an important proposal that, as the authors themselves describe, the result will be able to prepare health services for future local epidemics. The study may also benefit other locations, as declines in RSV cases followed increases have been documented in other nations during the Covid-19 pandemic.

I thoroughly enjoyed reviewing this manuscript and only have some minor requests for revision.

The methodology is compatible with the objective of the study. However, readers may benefit from additional information on i) location, general population and number of children under five (or older) years of age in Tokushima Prefecture; ii) type of access to the hospitals included in the study (public or private funding); iii) nominal or non-nominal identification of electronic medical records and iv) maximum age of children included in the study. Regarding the last item, I was unable to identify in the methods section up to what age the children were included in the study for the age group ≥24 months.

The hospitals included in the study are mentioned as the reference for the pediatric emergency care. Was there hospitalizations for non-resident children in Tokushima Prefecture at the reference hospitals during the study time? Was there reallocation of pediatric care to the reference hospitals in the study during the periods before and after the Covid-19 pandemic?

Did the electronic medical records allow for long-term follow-up of children (medical history)? Or, were the data about the variables obtaining only at the time of hospitalization for RSV? My question is fundamented in terms of the proportion of children ≥24 months of age with an unknown status regarding having received , palivizumab prophylaxis in comparison with children <6 months of age (estatistical significance). Could there be a memory bias regarding answering this question during hospitalization? Maybe, the information about the time and the measure procedures of the variable ", palivizumab prophylaxis" could offer more support for the discussion text about the differences of proportion.

Was there difference in the guidelines for rapid diagnostic testing and clinical diagnosis of RSV between the before and after periods? Was there a possibility of incorrect classification of the clinical diagnosis for other respiratory infections? Viral isolation was a widely used method for the diagnosis of RSV infection in the reference hospitals during the period of the study?

I would like to suggest that authors include tables of epidemiological changes in the characteristics of infants aged 6 to 11 and 12 to 23 months as a supplementary file.

6. PLOS authors have the option to publish the peer review history of their article (what does this mean?). If published, this will include your full peer review and any attached files.

**Do you want your identity to be public for this peer review?** For information about this choice, including consent withdrawal, please see our Privacy Policy.

Reviewer #1: **Yes: **Gabriel Berg de Almeida

Reviewer #2: No

---

## [Editor Report · Decision Letter 1]

12 May 2023

A surge in respiratory syncytial virus infection-related hospitalizations associated with the COVID-19 pandemic: An observational study at pediatric emergency referral hospitals in Tokushima Prefecture

PGPH-D-23-00225R1

Dear Mr. Shichijo,

We are pleased to inform you that your manuscript 'A surge in respiratory syncytial virus infection-related hospitalizations associated with the COVID-19 pandemic: An observational study at pediatric emergency referral hospitals in Tokushima Prefecture' has been provisionally accepted for publication in PLOS Global Public Health.

Best regards,

Everton Falcão de Oliveira, Ph.D

Academic Editor